# Phenolic Composition and Antioxidant Activity of *Alchemilla* Species

**DOI:** 10.3390/plants11202709

**Published:** 2022-10-13

**Authors:** Sebastian Kanak, Barbara Krzemińska, Rafał Celiński, Magdalena Bakalczuk, Katarzyna Dos Santos Szewczyk

**Affiliations:** 1Department of Pharmaceutical Botany, Medical University of Lublin, 1 Chodźki Str., 20-093 Lublin, Poland; 2Department of Cardiology, Independent Public Provincial Specialist Hospital in Chełm, 22-100 Chełm, Poland; 3Independent Unit of Functional Masticatory Disorders, Medical University of Lublin, 6 Chodźki Str., 20-093 Lublin, Poland

**Keywords:** *Alchemilla*, Rosaceae, polyphenols, flavonoids, tannins, antioxidant activity

## Abstract

The genus *Alchemilla*, belonging to the Rosaceae family, is a rich source of interesting secondary metabolites, including mainly flavonoids, tannins, and phenolic acids, which display a variety of biological activities, such as anti-inflammatory, antimicrobial, and antioxidant. *Alchemilla* species are used in traditional medicine for treatment of acute diarrhea, wounds, dysmenorrhea, and menorrhagia. In this review, we focus on the phenolic compound composition and antioxidative activity of *Alchemilla* species. We can assume that phytomedicine and natural products chemistry are of significant importance due to the fact that extract combinations with various bioactive compounds possess the activity to protect the human body rather than disturb damaging factors.

## 1. Introduction

*Alchemilla* is commonly called “Lady’s Mantle” or “lion’s foot” and representatives of this genus occur mostly across Europe and Asia which include northeastern Anatolia (Turkey), north of Iraq, and northwest of Iran. More than 300 species have been described from Europe, where large mountain ranges such as the Caucasus, the Alps, the Carpathians, and others with many endemic species are probably their main distribution centers [1].

The genus *Alchemilla* L. (F. Rosaceae Juss., subfam. Rosoidae Focke) includes a large number of forms that are not easy to identify. All *Alchemilla* species were considered for many years as one species—*Alchemilla vulgaris* L., within which at most a few vaguely defined varieties were distinguished. *Alchemilla* species are apparently very similar to each other, but apart from morphological diversity, they differ greatly in ecological, phytosociological, and geographic terms.

One of the earliest mentions of the *Alchemilla* species in scientific literature is dated to 1929 and was published by Harvard University Herbaria [2]. As reported by Ergene and co-authors [3], the *Alchemilla* genus consists of approximately 1000 species. Furthermore, according to The Plant List [4], 1722 plant name records match the search criteria for *Alchemilla*.

It has been reported that in Turkish folk medicine, species belonging to the *Alchemilla* are known locally as fındık out or aslan pençesi [5]. As reported by Afshar et al. [6], 24 species have been discovered within Iran and 14 among them are known to be endemic.

Aerial parts of miscellaneous *Alchemilla* species (Figure 1) are known to be excellent healing agents towards asthma, bronchitis, cough, and disorders connected with skin and liver inflammation [5].

Because of the fact that *Alchemilla vulgaris* possesses significant anti-inflammatory as well as astringent properties, it is a valued remedy for such ailments as ulcers, eczema, and menstrual disorders [7,8,9].

Moreover, *Alchemilla* species diminish symptoms of sore throat and alleviate nausea and vomiting [10]. *Alchemilla* species have been reported to possess a wide variety of biological activities, such as antioxidant, antibacterial, antiviral, anti-inflammatory, and ability to heal wounds in rats [11]. European Pharmacopoeia 6.0 describes *Alchemillae herba* as a medicinal agent with a variety of pharmacodynamics properties [12].

Among *Alchemilla* species that are the most widely researched for antimicrobial and antioxidant properties, *A. vulgaris*, *A. xanthochlora*, *A. diademata*, *A. rizeensis*, and *A. mollis* can be distinguished [13,14].

Various studies showed that *Alchemilla* species include miscellaneous compounds such as terpenes, hydrocarbons, fatty acids, and their esters as well as aldehydes, responsible for their pharmacological activities [15].

Other active compounds responsible for antioxidant and antimicrobial activities are tannins (composed of gallic and ellagic acid) and flavonoids (quercetin, luteolin, and proanthocyanidins) [16].

Due to the pharmaceutical and cosmetic importance of some *Alchemilla* species, in the present review, phenolic compounds occurring in the genus and antioxidant activity is discussed.

## 2. Methodology of Evidence Acquisition

This review focuses primarily on the content of phenolic compounds as well as antioxidant activity within plants belonging to the *Alchemilla* genus. All relevant literature databases were searched up to 19 June 2022. The database Google Scholar was used for searching articles with definite search terms, namely: *Alchemilla* phenolic compounds, *Alchemilla* polyphenols, *Alchemilla* antioxidant activity. Moreover, evidence was acquired with the use of the database Pubmed. Publications were identified using the search terms *Alchemilla* (all fields) and phenolic (all fields) or antioxidant (all fields). Articles that focused on and discussed the phenolic compounds and antioxidant activity of *Alchemilla* species, dating up to June 2022, were selected.

This review was done to highlight the diversity of phenolic compounds in the *Alchemilla* genus, and thus to reveal its healing potential. Furthermore, the aim of this paper was to provide up-to-date information on antioxidant activity of *Alchemilla* species, available in the scientific literature. Finally, we decided to discuss occurring knowledge gaps and propose recommendations concerning future research directions.

To the best of our knowledge, such a review has previously not been undertaken, thus the aim of the present study was to fill this knowledge gap.

## 3. Phenolic Compounds in the *Alchemilla* Species

Phenolic compounds are the most widely distributed secondary metabolites, ubiquitously present in the plant kingdom. Higher plants synthesize several thousand different phenolic compounds. The leaves contain, among others, amides and glycosides of hydroxycinnamic acids, esters, glycosylated flavonoids, and proanthocyanins and their relatives. Some soluble phenolics are widely distributed, but the distribution of many others is restricted to specific genera or families, making them easy biomarkers for taxonomic studies [17,18].

According to Choi and co-authors, plants from the genus *Alchemilla*, like other representatives of the Rosaceae family, are rich in polyphenols which are responsible for various pharmacological activities [7].

Another noteworthy observation is that various studies revealed the presence of phenolic compounds within the aerial parts of *Alchemilla* species, namely tannins, phenolic acids, flavonoids, and coumarins [6,19]. 

Flavonoids are an exceptionally large group of natural products (over 8000) that are found in many plant tissues, present inside the cells or on the surfaces of different plant organs [17].

Phytochemical investigations of *Alchemilla* species have led to the isolation diverse types of flavonoids, represented mostly by flavonols and flavones. Their structures are summarized in Figure 2, Figure 3, Figure 4, Figure 5, Figure 6, Figure 7, Figure 8, Figure 9, Figure 10 and Figure 11 and Table 1, Table 2, Table 3, Table 4, Table 5 and Table 6.

D’Agostino et al. (1998) isolated four flavonoid glycosides, namely 3-*O*-kaempferol-6”-*O*-(*p*-coumaroyl)-*β*-D-glucopyranoside (**7**), quercetin-3-*O*-*β*-D-rutinoside (**11**), quercetin-3-*O*-*β*-D-glucopyranoside (**14**), and quercetin-3-*O*-*α*-D-arabinofuranoside (**21**) from the methanol extract of *Alchemilla vulgaris* L. (Campania region, Italy) [20].

From the aqueous methanolic extract from the leaves of *A. speciosa* (Germany) astragalin (**2**), kaempferol 3-*O*-*β*-(2″-*O*-α-L-rhamnopyranosyl)-glucopyranoside uronic acid (**5**), kaempferol 3-*O*-*β*-D-glucuronide (**6**), **7**, **11**, miquelianin (**12**), **14**, quercitrin (**15**), quercetin 3-*O*-*β*-(2″-*O*-*α*-L-rhamnopyranosyl)-glucopyranoside uronic acid (**18**), hyperin (=hyperoside) (**13**), quercetin 3-*O*-*β*-D-sambubioside (**19**), quercetin 3-*O*-*β*-*ʋ*-sambubioside-7-*O*-*β*-D-glucoside (**20**), cynaroside (**26**), and scolymoside (**27**) were isolated [21]. 

Kaya et al. used TLC and HPLC techniques to separation of **11**, **13**, **14**, **15**, orientin (**25**), and vitexin (**32**) in 50% aqueous ethanol extracts from leaves of *A. hirtipedicellata*, *A. procerrima*, *A. sericata*, and *A. stricta* collected in the northeastern Black Sea region of Turkey [22]. The same authors identified **11**, **13–15**, **25**, and **32** in the extracts from the aerial parts of *A. bursensis*, *A. cimilensis*, *A. hirsutiflora*, *A. ikizdereensis, A. orduensis*, and *A. oriturcica* [23].

Lamaison and co-authors isolated miquelianin (**12**) from the aerial parts of *A. xanthochlora* [24]. This glucuronide of quercetin was later found also in the aerial parts of *A. barbatiflora* [25], *A. caucasica* [10], *A. achtarowii* [26], *A. mollis* [27], *A. persica* [6], *A. vulgaris* [28], *A. coriacea*, *A. filicaulis*, and *A. glabra* [29]. 

From the aerial flowering parts of *A. mollis* (Bulgaria), **7**, **13**, **14**, rhodiolgin (**36**, Figure 4), gossypetin-3-*O*-*β*-D-galactopyranosyl-7-*O*-*α*-L-rhamnopyranoside (**37**, MW 626), and sinocrassoside D2 (**45**, MW 626) were isolated [27]. One year later, Trendafilova et al. isolated astragalin (**2**), kaempferol 3-*O*-(4”-*E*-*p*-coumaroyl)-robinobioside (variabiloside G, **3**), **7**, quercetin-3-*O*-α-D-arabinopyranoside (**10**), **13**, and **14** from the aerial parts of *A. achtarowii* [26].

Phytochemical studies on active fractions of the water subextract led to the isolation of kaempferol-3-*O*-*β*-D-xylopyranoside (**4**), **7**, **10**, **12**, **13**, and **34** from the aerial parts of *A. barbatiflora* (Turkey) [25]. Guaijaverin (**10**) was also identified in the aerial parts of *A. xanthochlora* (France) [30].

Neagu et al. applied liquid chromatography coupled to tandem mass spectrometry to identify polyphenolic compounds extracted by water and aqueous ethanol (70% *v*/*v*) from *A. vulgaris*. Flavonols (kaempferol **1**, quercetin **9**, **11**, myricetin **38**, Figure 5), flavanols (**35**), flavones (luteolin **24**, Figure 6), isoflavones (genistein **39**, daidzein **40**, Figure 7), and flavonoid glucosides (**14**) were detected in the plant samples [31].

Using high performance liquid chromatography (HPLC) analysis, Denev et al. (2014) found that aerial parts of *A. glabra* (Plovdiv, Bulgaria) contained **11**, **34**, and **35** [32]. Akkol and co-authors used the same analytical method to identify **13** and **14** in the aerial parts of *A. mollis* and *A. persica* (Turkey) [33].

Duckstein et al. investigated acetone/water extracts from the leaves, including stalks, of *A. vulgaris* and *A. mollis* (Germany) for their phenolic composition by liquid chromatography-tandem mass spectrometry (LC-MS/MS). Compounds **12**, methyl-quercetin glucuronide (**16**), quercetin hexoside (**17**), quercetin-feruloyl hexose (**22**), and quercetin hexoside-deoxyhexoside (**23**) were reported [34]. 

Compounds **1**, **9**, **11**, and **34** were found in the ethanol extracts from the aerial parts of *A. vulgaris* (Russia) [35]. Karatoprak et al. reported that different (hexane, ethyl acetate, methanol, butanol) extracts from the aerial parts of *A. mollis* (Turkey) contained **11**, **26**, cosmosiin (**33**), **34**, and **35** (Figure 8) [36]. 

Twenty flavonoids were identified in aqueous ethanol (80% *v*/*v*) from the leaves of *A. vulgaris* (Egypt). Among them, **1**, **9**, **11**, **24**, **31** (Figure 9), **33**, **34**, **35**, naringenin (**43**), luteolin 6-arabinose 8-glucose (**46**, MW 610), luteolin 6-glucose 8-arabinose (**47**, MW 610), apigenin 6-arabinose 8-galactose (**48**), apigenin 6-rhamnose 8-glucose (**49**), apigenin 7-*O*-neohespiroside (**50**, MW 578; Figure 10), kaempferol 3,7-dirhamoside (**51**), hesperetin (**52**), kaempferol 3-(2-*p*-comaroyl)glucose (**53**), and rhamnetin (**54**) were determined [37].

In the aerial parts of *A. vulgaris*, **1**, **2**, **7**, **9**, **11**, **13**, **14**, **15**, **24**, **26**, **31**, **33**, **34**, **39**, morin (**41**, MW 302), chrysoeriol (**42**), and **43** were reported [38,39]. Afshar et al. used HR-MS Q-TOF for the identification of nicotiflorin (**8**), catechin (**34**), epicatechin (**35**), and aromadendrin glucoside (**55**) in the aerial parts of *A. persica* (Eastern Azarbaijan) [6]. Catechin is regarded as one of the most powerful antioxidants. Moreover, some research has shown the a strong relationship between this compound and inhibition of carcinogenesis (e.g., breast and ovarian cancer cell growth) [39].

In the latest study from 2022, Dos Santos Szewczyk et al. reported that the aerial parts and roots of *A. acutiloba* Opiz (Poland) contained **1**, **2**, **8**, **9**, **11**, **13**, **14**, **15**, **24**, isorhamnetin (**28**), isorhamnetin-3-glucoside (**29**), narcissoside (**30**), and eriodictyol (**44**, Figure 11) [40].

Recent studies have revealed the presence of such flavonoids as **2**, **7**, **11–14**, and **16** in methanol extracts from the aerial parts of *A. viridiflora* (Bulgaria) [41]. 

The phenolic compounds identified in the *Alchemilla* genus are summarized in Table 7.

The plant tannins are a unique group of phenolics of relatively high molecular weight with the ability to complex strongly with carbohydrates and proteins. In higher plants, tannins consist of two major groups of metabolites: the hydrolysable tannins and condensed tannins [17]. *Alchemilla* species as members of Rosaceae family also produce, apart from flavonoids, tannins.

Geiger et al. identified in the aerial parts of *A. xanthochlora* (Germany) ellagitannins such as agrimoniin (**58**, MW 1871), pedunculagin (**59**, MW 784)), and laevigatin F (**60**, MW 802) [42].

Duckstein and co-authors used liquid chromatography-tandem mass spectrometry (LC-MS/MS) to identify tannins extracted by acetone/water (8/2, *v*/*v*) from the leaves (including stalks) from *A. vulgaris* and *A. mollis* (Germany). Compounds **58**, **59**, castalagin/vescalagin isomer (**61**, MW 934), galloyl-HHDP hexose (**62**, MW 618), trigalloyl hexose (**63**, MW 636), and sanguiin (**64**, MW 1871) were detected in studied plant samples [34].

In the aerial parts of *A. mollis* [36] and *A. persica* [6], methyl gallate (**65**, Figure 12) was identified. The authors [6] found also that extracts of *A. persica* contained **58**, **59**, **62**, **64**, casuarictin (**66**, Figure 13), and digalloyl-galloyl galloside (**67**, MW 1084). 

Recent studies have revealed the presence in the aerial parts of *A. viridiflora* such tannins as HHDP-hexoside (**68**), brevifolincarboxylic acid (**69**, Figure 14), and tellimagrandin I (**70**, Figure 15) and II (**71**, MW 938) which have been reported for the first time in *Alchemilla* species [41].

Phenolic acids are also common in higher plants, and they are usually present in the bound soluble form conjugated with sugars or organic acids [17] (Figure 16 and Figure 17, Table 8 and Table 9).

In the aqueous extracts and ethanolic extracts (70% (*v*/*v*) ethanol) of *A. vulgaris* caffeic acid (**73**), chlorogenic acid (**74**, Figure 18), elagic acid (**77**), ferulic acid (**78**), gallic acid (**79**), *p*-coumaric acid (**82**), rosmarinic acid (**87**, Figure 19), and sinapic acid (**89**) were identified using HPLC-MS analysis [31]. 

Denev et al. reported that the aerial parts of *A. glabra* contained **73**, **74**, 3,4-dihydroxybenzoic acid (**76**), and **79** [32]. In the leaves, including stalks, of *A. vulgaris* and *A. mollis*, three phenolic acids (**74**, **77**, **79**) were found [34]. Moreover, in different extracts from the aerial parts of *A. mollis*, **73**, **79**, and gentisic acid (**80**) were noticed [36]. 

In the 80% methanol extracts from the leaves of *A. jumrukczalica* and *A. vulgaris* (Bulgaria), free and bonded phenolic acids were identified. Among reported phenolic acids, **73**, **80**, protocatechuic (**81**), salicylic (**88**), *trans*-cinnamic (**91**), and vanilic (**93**) acids were the major compounds [43].

Fifteen phenolic acids (benzoic acid (**72**), **73**, **74**, **78**, **79**, **81**, **82**, 4-hydroxybenzoic acid (**83**), 3,4,5-methoxycinnamic acid (**85**), **87**, **88**, **91**, **93**) in the leaves [37] and **77** [38], and **73, 74**, 2,5-dihydroxybenzoic acid (**75**), **78**, **79**, **81–83**, as well as quinic acid (**94**, Figure 20) [39] were found in the aerial parts of *A. vulgaris*. The advantages of identified phenolic acids are associated with several health benefits such as antioxidant, anti-diabetic, and anticancer effects [39].

Moreover, Dos Santos Szewczyk et al. [40] reported that aerial parts and roots of *A. acutiloba* contained **73**, **78–83**, **87**, **88**, syringic acid (**90**), and **93**.

## 4. Antioxidant Activity

It has been proven that oxidative stress participates in the formation of various diseases such as chronic obstructive pulmonary disease (COPD), Alzheimer’s disease, atherosclerosis, and cancer [46].

The evidence described above confirms the importance of searching for new effective and safe antioxidant agents. As reported by Forman and Zhang [46], two major mechanisms connected with diseases formation which contribute to cellular damage can be distinguished, namely: generation of reactive oxygen species (•OH, ONOO−, HOCl) which directly oxidize macromolecules, especially membrane lipids, enzymes, proteins, as well as nucleic acids, and leads to death resulting from aberrant cell function. Furthermore, the second pathway relates with aberrant redox signaling.

Based on all the above, in this review, we sought to clarify in more detail the antioxidant potential of *Alchemilla* species (Table 10).

Ondrejovič et al. evaluated the antioxidant activities of various solvent extracts and fractions acquired by solid-liquid and liquid-liquid extraction and column chromatography from leaves of *A. xanthochlora*. The most prominent antioxidant activity was observed in methanolic extract. Isolated fraction showed antioxidant activity of 535.2 mg DPPH per g of fraction residue [13]. 

Antioxidant activity of the extracts from the aerial parts and roots of *A. persica* (Turkey) was evaluated using the DPPH radical scavenging assay and measurement of malondialdehyde (MDA) levels. The extracts were found to exhibit DPPH (1,1-diphenyl-2-picrylhydrazyl) free radical scavenging activity with IC_50_ values of 0.055 M and 0.151 M for the aerial parts and roots, respectively. The MDA level of the aerial parts was found to be 5.9 nmol/mL, and 19.08 nmol/mL for the roots [3].

Nikolova et al. evaluated antioxidant activity of the methanol extracts from leaves of *A. jumrukczalica* and *A. vulgaris* by the scavenging effect on DPPH radicals. The extracts showed good antiradical activity with IC_50_ values of 12.09 μg/mL and 19.62 μg/mL, respectively. Obtained values were comparable with those of butylated hydroxytoluene (BHT)—12.65 μg/mL and syringic acid—4.40 μg/mL, used as standard substances [43].

Boroja and co-authors [58] analyzed antioxidant activity of the methanol extract from aerial parts of *A. vulgaris*. They found that studied extract possessed DPPH inhibition activity with IC_50_ = 5.40 µg/mL. Moreover, the ABTS results demonstrated IC_50_ value of 60.10 µg/mL. The same authors evaluated antioxidant efficacy of extracts from the roots and aerial parts of *A. vulgaris* (Central Serbia) as total antioxidant capacity, metal chelation and reducing power ability, inhibition of lipid peroxidation, as well as their potential to neutralize DPPH, ABTS, and OH radicals. They found that roots exert a higher total antioxidant activity than aerial parts (316.5 and 265.6 mg ascorbic acid/g, respectively). Comparable results for both extracts were obtained in the reducing power assay (633.0—aerial parts and 607.5 mg Trolox/g—roots). In the ferrous ion chelating test, all studied samples failed to chelate Fe^2+^ at concentration 1 mg/mL [11].

Antioxidant capacity of the methanolic extract and fractions of *A. mollis* was also measured by their ability to scavenge the DPPH radical. The EtOAc fraction was found to be the most active radical scavenger (IC_50_ = 9.8 ± 1.8 μg/mL) but this value was less than that of quercetin (IC_50_ = 3.2 ± 0.4 μg/mL) [27]. The antioxidant capacity of stalks of *A. mollis* aqueous ethanol extracts was also investigated. The activity was determined by four different assays (FRAP (ferric-reducing antioxidant power), CUPRAC (cupric ion reducing antioxidant capacity), DPPH, and ABTS) and was expressed as mmol Trolox equivalent per dm^3^ extract. The maximum values of extracts were 382.78 ± 1.16; 363.79 ± 0.74; 247.58 ± 2.26; and 308.44 ± 6.74 for FRAP, CUPRAC, DPPH, and ABTS assays, respectively [55]. Karatoprak et al. evaluated antioxidant activity of hexane, ethyl acetate, methanol, butanol, water, and 70% methanol extracts from the aerial parts of *A. mollis*. In the DPPH assay, the IC_50_ values were found to be 0.21 mg/mL and 0.24 mg/mL, respectively for 70% MeOH and water extracts. ABTS^+•^ radical scavenging effects of the extracts were studied at the doses of 0.25 and 0.5 mg/mL. All the extracts showed the highest level of activity at 0.5 mg/mL. The TEAC values of 70% methanol and water extracts (0.75 and 0.83 mM/Trolox) were found higher than the methanol, ethyl acetate, and hexane extracts [36]. The authors studied antiradical activity of various extracts from the herb of *A. mollis* also in different research. The IC_50_ values in the DPPH test were found 0.264 mg/mL, 0.146, and 0.161 mg/mL, respectively, for water, deodorized water, and 50% MeOH extracts. All extracts also managed to inhibit the ABTS^●+^ radical. The TEAC values of water extract (0.90 and 1.55 mM/L/Trolox) were found higher than the deodorized water and 50% MeOH extracts [54].

Denev et al. evaluated extracts of the aerial parts of *A. glabra* by means of several assays, including ORAC, TRAP, HORAC, and inhibition of lipid peroxidation. Between all extracts studied, *A. glabra* extract revealed the second highest chelating ability expressed as a HORAC value of 1999.4 μmol GAE/g [32].

Hamid and co-authors analyzed antioxidant activities of *A. vulgaris* roots using Trolox equivalent antioxidant capacity (TEAC), ferric reducing antioxidant power (FRAP), and TBARS (thiobarbituric acid reactive substances) assays. The antioxidant activity measured with TEAC was 68.21 mmol of Trolox (TE)/g of dry weight (DW). Whereas FRAP assay was 40.12 mmol of TE/g DW [59].

Vlaisavljević et al. evaluated the various extracts (80% methanol, 70% ethanol, 70% ethylacetate, and distilled water) of the aerial parts of *A. vulgaris* by means of different in vivo assays. The authors found that ethylacetate extract demonstrated the highest antioxidant potential, where the most pronounced and significant antioxidant effect of tested extracts was observed for DPPH and FRAP assays (DPPH: 502.56 mg TE per g extract; FRAP: 8745.31 mg EAA per g of dry extract), followed by methanol extract [39].

Antioxidant effects of the extracts, fractions, and isolated compounds from the aerial parts of *A. barbatiflora* were estimated using several methods as 1,1-diphenyl-2-picryl-hydrazyl (DPPH), superoxide radical scavenging (SOD), phosphomolibdenum-reducing antioxidant power (PRAP), and ferric-reducing antioxidant power (FRAP) assays. Crude methanol extract showed remarkable DPPH and SOD radical scavenging activities with 83.44% and 83.34% at 125 μg/mL. Among the tested sub-extracts, water sub-extract showed the best results with 83.06%, 96.08%, and 97.17% at 125, 250, and 500 μg/mL, respectively, for DPPH scavenging activities. Hexane sub-extract had moderate DPPH scavenging activities as compared to gallic acid. In SOD assay at 125 μg/mL, water sub-extract displayed significant SOD radical scavenging activities with 81.07%. Moreover, water sub-extract showed higher absorbance than methanol extract in PRAP assay. In FRAP assay, the result of methanol extract was found as 44.32 mg BHAE/g extract while the result of water sub-extract was as 93.46 mg BHAE/g extract [25].

The antioxidant activities of various extracts of the aerial parts and roots of *A. acutiloba* were determined using the DPPH^•^ and ABTS^•+^ radical scavenging assays. It was found that at a dose of 50.0 µg/mL, the DPPH^•^ scavenging abilities were the highest for the ethyl acetate (94.85%) and butanol (87.31%) fraction of the aerial parts, and in the ABTS^•+^ assay for butanol fraction (80.56%). Among studied extracts, butanol fraction of the aerial parts was also the most active ones interfering with the formation of iron and ferrozine complexes (IC_50_ = 11.43 µg/mL of DE) [40]. 

## 5. Conclusions and Research Gaps/Future Investigations

The available data suggest that recent times have brought a fundamental change in classical medicine considering the treatment of diseases. The above-mentioned phenomenon is related to large-scale application of drug combinations, thus the multidrug way of treatment is currently of great significance. Simultaneously, mono-substance therapies are becoming less and less popular. Moreover, a gradual development of drugs that activate natural defense and protective as well as repair mechanisms instead of impairing disadvantageous agents (such as cancer cells and microorganisms) can be observed [60].

Taking into consideration the above-mentioned facts, we can assume that phytomedicine and natural products chemistry are of significant importance due to the fact that extract combinations with various bioactive compounds can protect the human body rather that disturb damaging factors.

The World Health Organization (WHO) indicates that application of 74% of the curatives with plant origin using in modern medicine correlates with their traditional usages in various traditional medicines [10].

## Figures and Tables

**Figure 1 plants-11-02709-f001:**
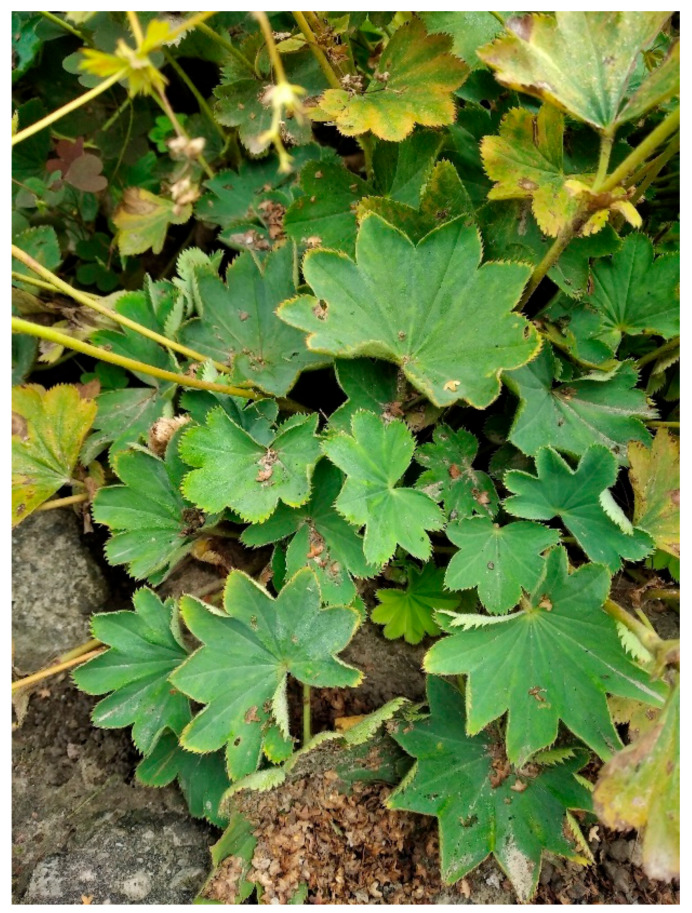
*Alchemilla peristerica* Pawł. aerial parts.

**Figure 2 plants-11-02709-f002:**
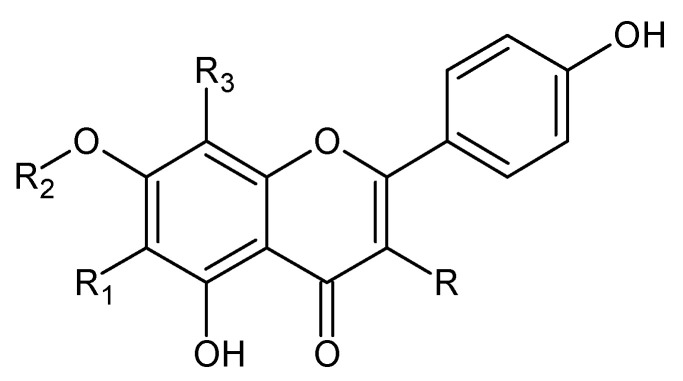
Kaempferol and kaempferol *O*-glycosides.

**Figure 3 plants-11-02709-f003:**
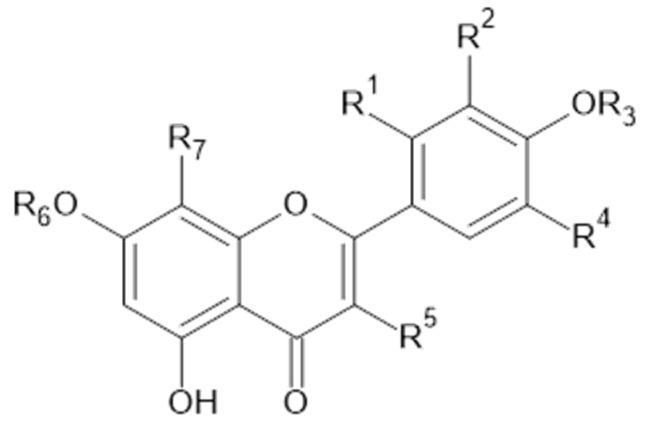
Quercetin and derivatives.

**Figure 4 plants-11-02709-f004:**
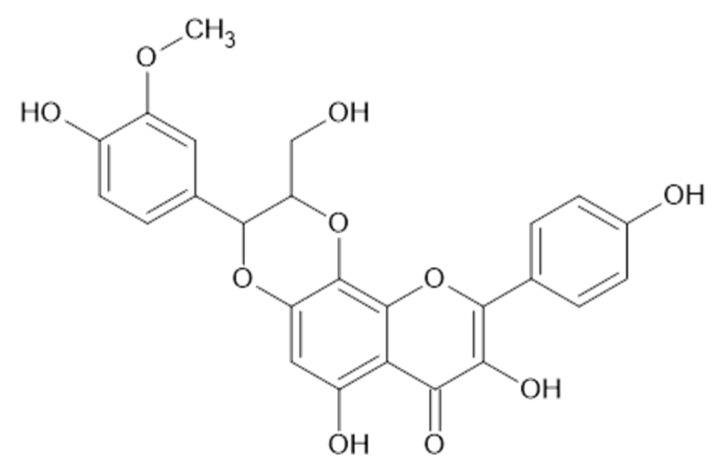
Rhodiolgin (**36**, MW 464).

**Figure 5 plants-11-02709-f005:**
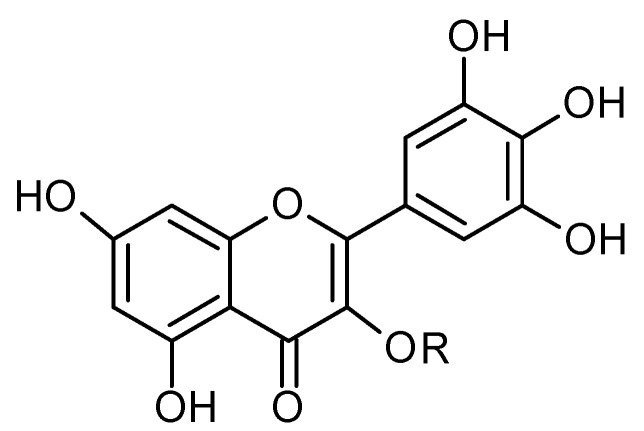
Chemical structure of compound **38** (R = H; MW = 318).

**Figure 6 plants-11-02709-f006:**
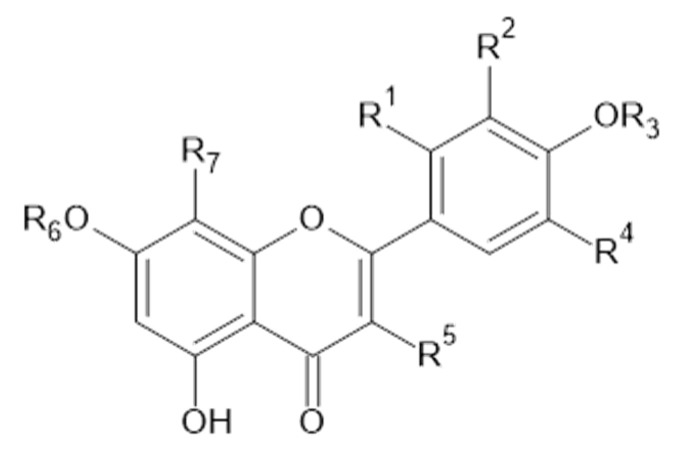
Luteolin and isorhamnetin, and derivatives.

**Figure 7 plants-11-02709-f007:**
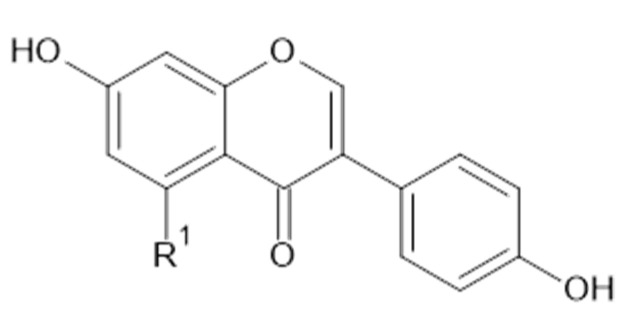
Chemical structure of compound **39** (R_1_ = OH; MW = 270) and **40** (R_1_ = H; MW = 254).

**Figure 8 plants-11-02709-f008:**
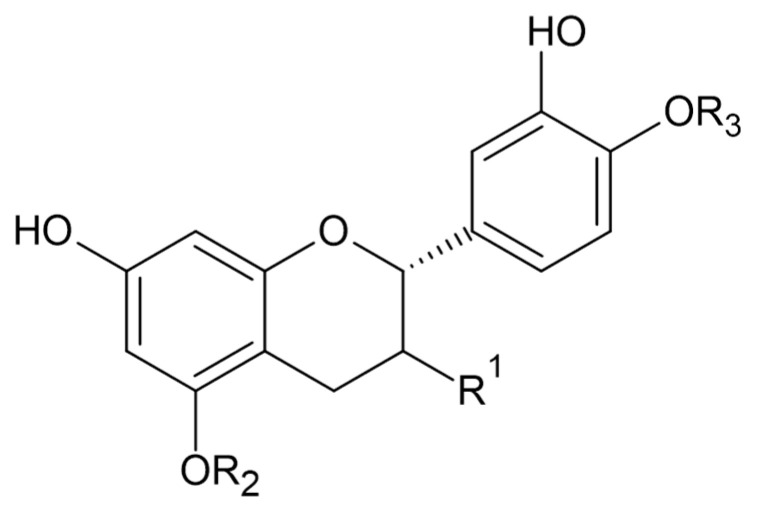
Catechin and epicatechin.

**Figure 9 plants-11-02709-f009:**
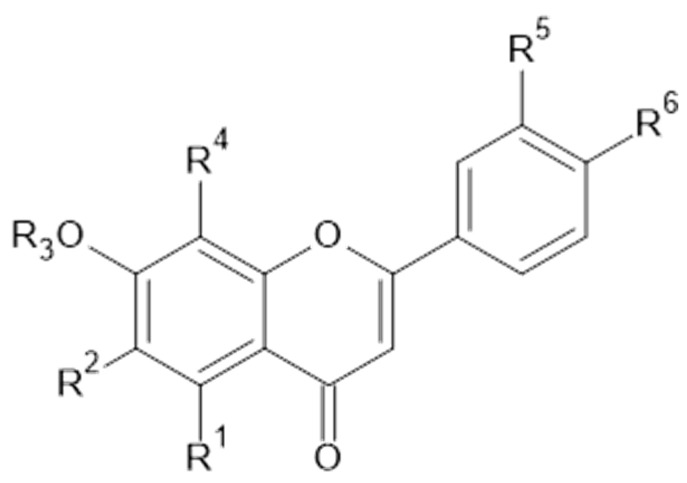
Apigenin and derivatives.

**Figure 10 plants-11-02709-f010:**
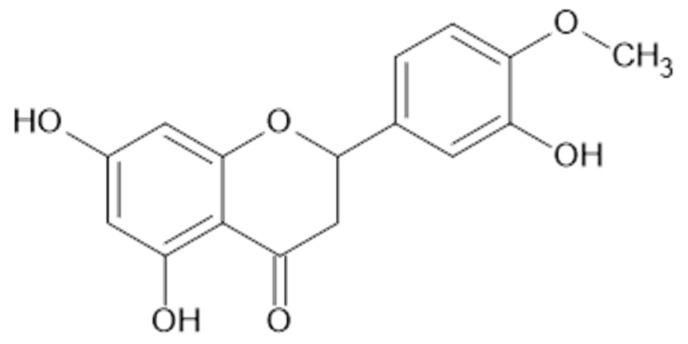
Chemical structure of compound **50** (MW = 302).

**Figure 11 plants-11-02709-f011:**
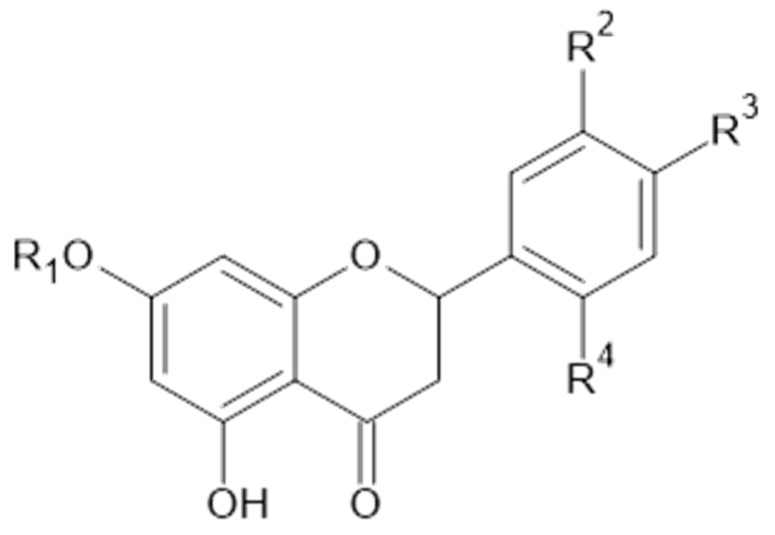
Chemical structure of compound **43** and **44**.

**Figure 12 plants-11-02709-f012:**
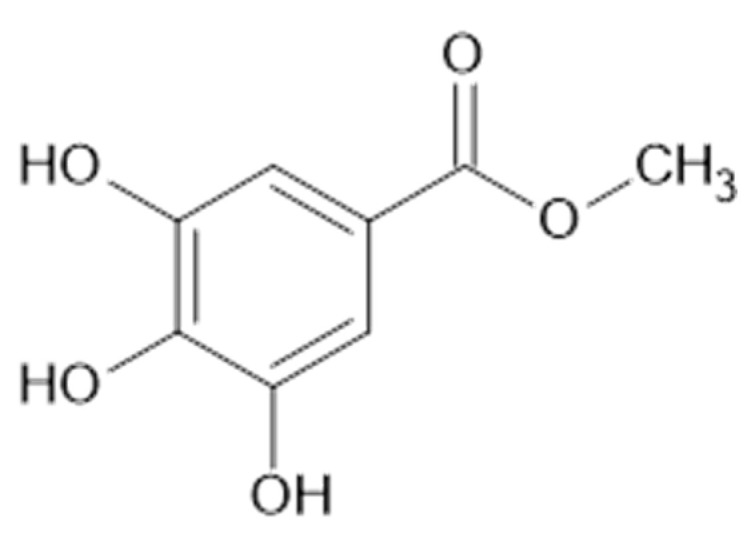
Chemical structure of compound **65** (MW = 184).

**Figure 13 plants-11-02709-f013:**
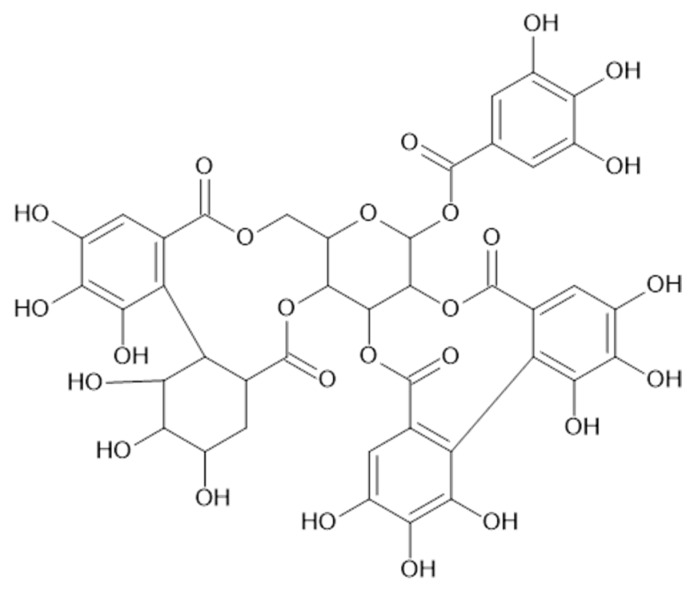
Chemical structure of compound **66** (MW = 936).

**Figure 14 plants-11-02709-f014:**
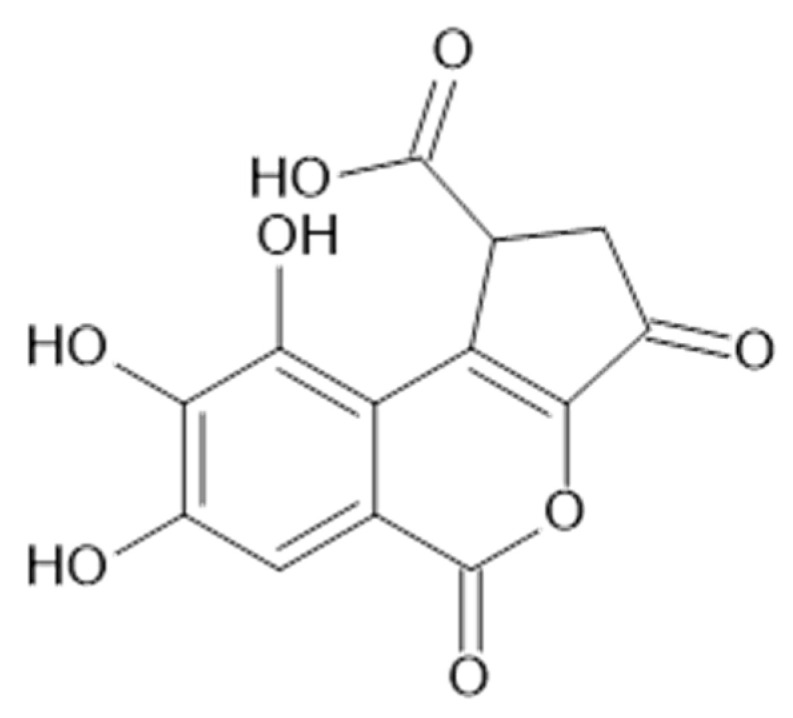
Chemical structure of compound **69** (MW = 292).

**Figure 15 plants-11-02709-f015:**
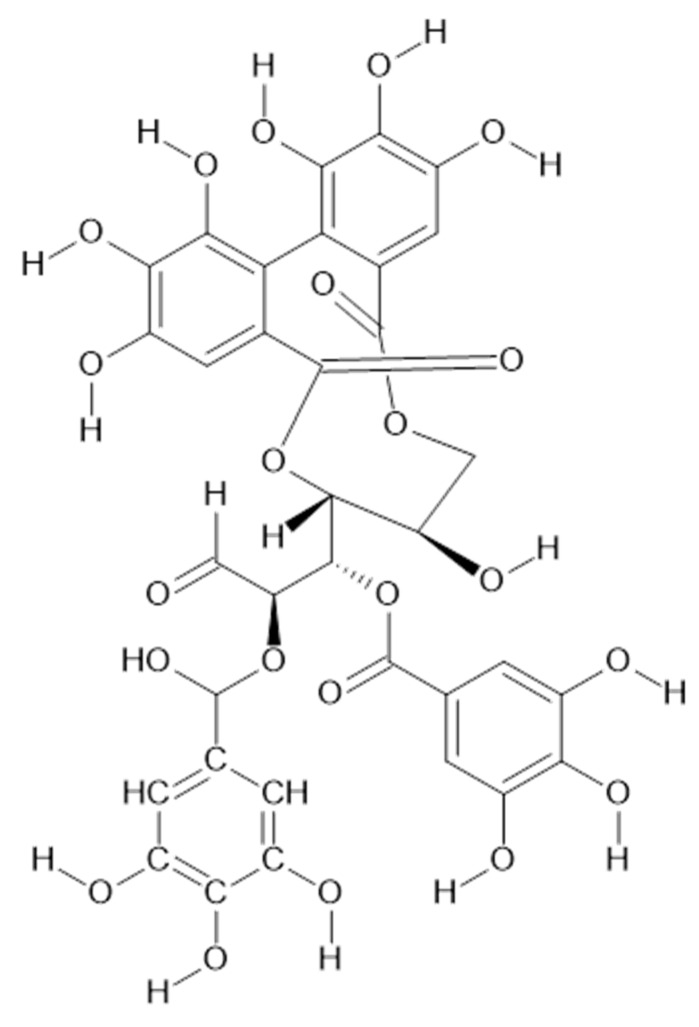
Chemical structure of compound **70** (MW = 786).

**Figure 16 plants-11-02709-f016:**
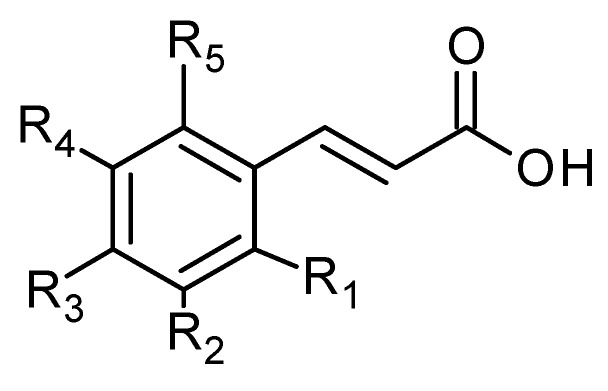
Hydroxycinnamic acid derivatives.

**Figure 17 plants-11-02709-f017:**
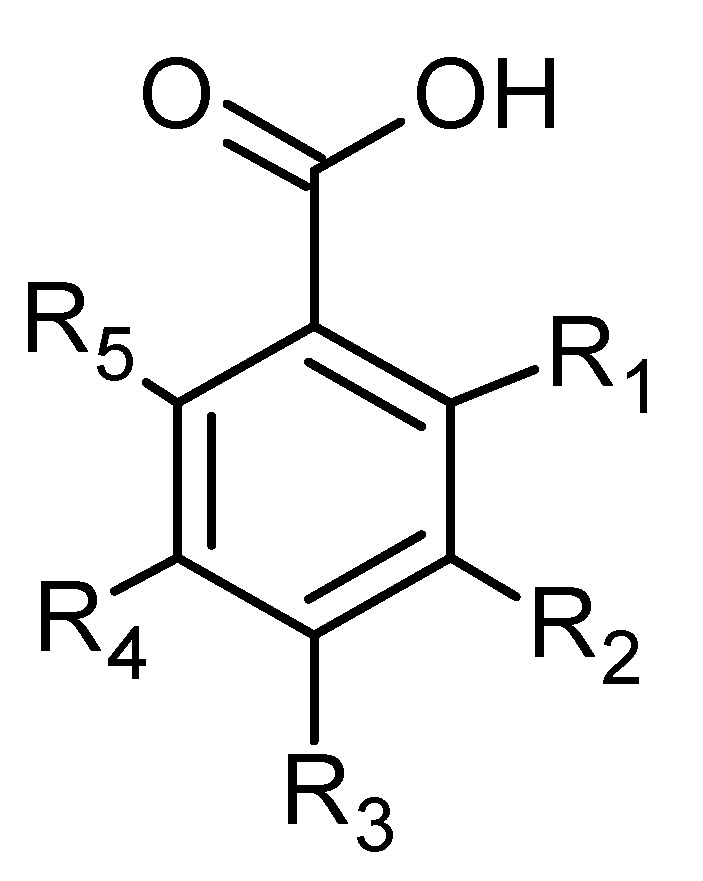
Benzoic acid derivatives.

**Figure 18 plants-11-02709-f018:**
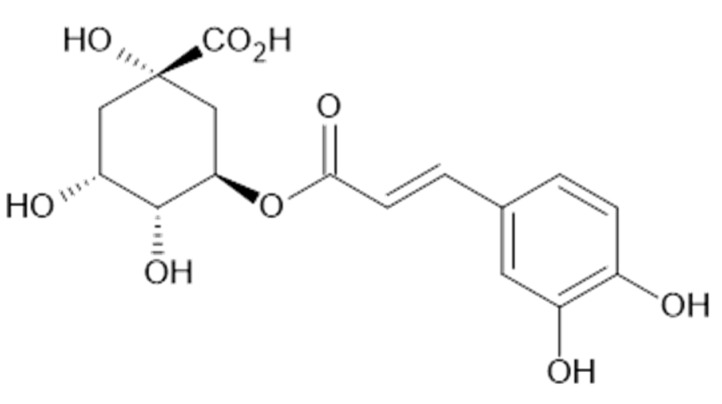
Chemical structure of compound **74** (MW = 354).

**Figure 19 plants-11-02709-f019:**
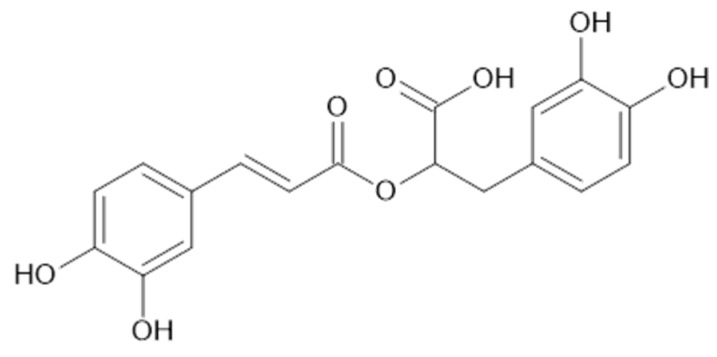
Chemical structure of compound **87** (MW = 360).

**Figure 20 plants-11-02709-f020:**
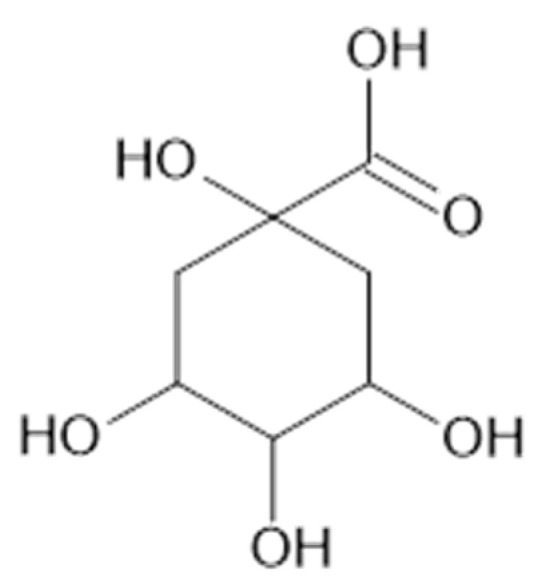
Chemical structure of compound **94** (MW = 192).

**Table 1 plants-11-02709-t001:** Kaempferol and kaempferol *O*-glycosides.

Compound	R	R_1_	R_2_	R_3_	MW (g/mol)
**1**	OH	H	H	H	286
**2**	*O*-glc	H	H	H	448
**3**	*p*-coumaroyl-robinobioside	H	H	H	918
**4**	xyl	H	H	H	418
**5**	2″-*O*-α-L-rha-β-D-glc	H	H	H	594
**6**	glcA	H	H	H	462
**7**	6″-*O*-(E)-*p*-coumaroyl)glc	H	H	H	594
**8**	*O*-rha-glc	H	H	H	594
**51**	*O*-rha	H	rha	H	578
**53**	2-*p*-coumaroyl-glc	H	H	H	594

**Table 2 plants-11-02709-t002:** Quercetin and derivatives.

Compound	R_1_	R_2_	R_3_	R_4_	R_5_	R_6_	R_7_	MW (g/mol)
**9**	H	OH	H	H	OH	H	H	302
**10**	H	H	H	OH	*O*-*α*-L-ara	H	H	434
**11**	H	OH	H	H	*O*- glc(6→1)rha	H	H	610
**12**	H	H	H	OH	*O*-glucuronide	H	H	478
**13**	H	OH	H	H	*O*- gal	H	H	464
**14**	H	OH	H	H	*O*- glc	H	H	464
**15**	H	OH	H	H	*O*- rha	H	H	448
**19**	H	OH	H	H	*O*-*β*-D-xyl-(2→1)- *β*-D-glc	H	H	596
**20**	H	OH	H	H	*O*-β-D-xyl-(2→1)- *β*-D-glc	glc	H	758
**21**	H	OH	H	H	*O*- *α*-D-ara-furanoside	H	H	434
**54**	H	OH	H	H	OH	CH_3_	H	316
**56**	H	OH	H	H	*O*-ara-furanoside	H	H	434

**Table 3 plants-11-02709-t003:** Luteolin and isorhamnetin, and derivatives.

Compound	R_1_	R_2_	R_3_	R_4_	R_5_	R_6_	R_7_	MW (g/mol)
**24**	H	OH	H	H	H	H	H	286
**25**	H	OH	H	H	H	H	C-*β*-D-glc	448
**26**	H	OH	H	H	H	*O*-*β*-D-glc	H	448
**27**	H	OH	H	H	H	rha-glc	H	594
**28**	H	OCH_3_	H	H	OH	H	H	316
**29**	H	OCH_3_	H	H	*O*-glc	H	H	478
**30**	H	OCH_3_	H	H	*O*-rha glc	H	H	624
**42**	H	OCH_3_	H	H	H	H	H	300

**Table 4 plants-11-02709-t004:** Catechin and epicatechin.

Compound	R_1_	R_2_	R_3_	MW (g/mol)
**34**	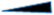 OH	H	OH	290
**35**	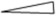 OH	H	OH	290

**Table 5 plants-11-02709-t005:** Apigenin and derivatives.

Compound	R_1_	R_2_	R_3_	R_4_	R_5_	R_6_	MW (g/mol)
**31**	OH	H	H	H	H	OH	270
**32**	OH	H	H	glc	H	OH	432
**33**	OH	H	glc	H	H	OH	432
**57**	OH	H	H	H	H	OCH_3_	284

**Table 6 plants-11-02709-t006:** Chemical structure of compound **43** and **44**.

Compound	R_1_	R_2_	R_3_	R_4_	MW (g/mol)
**43**	H	H	OH	H	272
**44**	H	OH	OH	H	288

**Table 7 plants-11-02709-t007:** The overview on the phenolic compounds identified in the *Alchemilla* genus.

Constituent Name	Species	Part of Plant	References
**1.** Kaempferol	*A. acutiloba*	aerial parts, roots	[40]
	*A. vulgaris*	aerial parts	[31,35,37,38,39]
2. Astragalin	*A. acutiloba*	aerial parts, roots	[40]
	*A. achtarowii*	aerial parts	[26]
	*A. speciosa*	leaves	[21]
	*A. viridiflora*	aerial parts	[41]
	*A. vulgaris*	aerial parts	[38,39]
**3.** Variabiloside G	*A. achtarowii*	aerial parts	[26]
**4.** Kaempferol-3-*O*-*β*-D-xylopyranoside	*A. barbatiflora*	aerial parts	[25]
**5.** Kaempferol 3-*O*-*β*-(2″-*O*-α-L-rhamnopyranosyl)-glucopyranoside uronic acid	*A. speciosa*	leaves	[21]
**6.** Kaempferol 3-*O*-*β*-D-glucuronide	*A. speciosa*	leaves	[21]
**7.** Kaempferol 3-*O*-*β*-D-(6”-*O*-(*E*)-*p*-coumaroyl) glucopyranoside	*A. achtarowii*	aerial parts	[26]
	*A. barbatiflora*	aerial parts	[25]
	*A. mollis*	aerial parts	[27]
	*A. speciosa*	aerial parts	[21]
	*A. viridiflora*	aerial parts	[41]
	*A. vulgaris*	aerial parts	[20,38]
**8.** Nicotiflorin	*A. acutiloba*	aerial parts, roots	[40]
	*A. persica*	aerial parts	[6]
**9.** Quercetin	*A. acutiloba*	aerial parts, roots	[40]
	*A. vulgaris*	aerial parts leaves	[31,35,37,38,39]
**10.** Guaijaverin	*A. achtarowii*	aerial parts	[26]
	*A. barbatiflora*	aerial parts	[25]
	*A. xanthochlora*	aerial parts	[30]
**11.** Rutin	*A. acutiloba*	aerial parts	[40]
	*A. hirtipedicellata* *A. procerrima* *A. sericata* *A. stricta*	leaves	[22]
	*A. glabra*	aerial parts	[32]
	*A. bursensis* *A. cimilensis* *A. hirsutiflora* *A. ikizdereensis* *A. orduensis* *A. oriturcica*	aerial parts	[23]
	*A. mollis*	aerial parts	[27,36]
	*A. speciosa*	aerial parts	[21]
	*A. viridiflora*	aerial parts	[41]
	*A. vulgaris*	aerial parts	[20,31,35,37,38,39]
**12.** Miquelianin	*A. barbatiflora*	aerial parts	[25]
	*A. caucasica*	aerial parts	[10]
	*A. achtarowii*	aerial parts	[26]
	*A. mollis*	aerial parts	[27]
	*A. persica*	aerial parts	[6]
	*A. speciosa*	aerial parts	[21]
	*A. viridiflora*	aerial parts	[41]
	*A. vulgaris*	aerial parts	[28]
	*A. xanthochlora*	aerial parts	[24]
**13.** Hyperoside	*A. achtarowii*	aerial parts	[26]
	*A. acutiloba*	aerial parts, roots	[40]
	*A. hirtipedicellata* *A. procerrima* *A. sericata* *A. stricta*	leaves	[22]
	*A. barbatiflora*	aerial parts	[25]
	*A. coriacea* *A. filicaulis* *A. glabra*	aerial parts	[29]
	*A. armeniaca* *A. bursensis* *A. cimilensis* *A. hirsutiflora* *A. ikizdereensis* *A. orduensis* *A. oriturcica*	aerial parts	[23]
	*A. mollis*	aerial parts	[27,33]
	*A. persica*	aerial parts	[33]
	*A. speciosa*	leaves	[21]
	*A. viridiflora*	aerial parts	[41]
	*A. vulgaris*	aerial parts	[38]
**14.** Isoquercitrin	*A. achtarowii*	aerial parts	[26]
	*A. acutiloba*	aerial parts, roots	[40]
	*A. hirtipedicellata* *A. procerrima* *A. sericata* *A. stricta*	leaves	[22]
	*A. bursensis* *A. cimilensis* *A. erzincanensis* *A. orduensis* *A. oriturcica*	aerial parts	[23]
	*A. mollis*	aerial parts	[27,33]
	*A. persica*	aerial parts	[33]
	*A. speciosa*	aerial parts	[21]
	*A. viridiflora*	aerial parts	[41]
	*A. vulgaris*	aerial parts	[20,31,38]
**15.** Quercitrin	*A. acutiloba*	aerial parts, roots	[40]
	*A. hirtipedicellata* *A. procerrima* *A. sericata* *A. stricta*	leaves	[22]
	*A. hirsutiflora* *A. orduensis*	aerial parts	[23]
	*A. speciosa*	aerial parts	[21]
	*A. vulgaris*	aerial parts	[39]
**16.** Methyl-quercetin glucuronide	*A. mollis*	leaves	[34]
	*A. viridiflora*	aerial parts	[41]
**17.** Quercetin hexoside	*A. mollis*	leaves	[34]
**18.** Quercetin 3-*O*-*β*-(2″-*O*-*α*-L-rhamnopyranosyl)-glucopyranoside uronic acid	*A. speciosa*	leaves	[21]
**19.** Quercetin 3-*O*-*β*-D-sambubioside	*A. speciosa*	leaves	[21]
**20.** Quercetin 3-*O*-*β*-*ʋ*-sambubioside-7-*O*-*β*-D-glucoside	*A. speciosa*	leaves	[21]
**21.** Quercetin-3-*O*-*α*-D-arabinofuranoside	*A. vulgaris*	aerial parts	[20]
**22.** Quercetin-feruloyl hexose	*A. vulgaris*	leaves	[34]
**23.** Quercetin hexoside-deoxyhexoside	*A. vulgaris*	leaves	[34]
**24.** Luteolin	*A. acutiloba*	aerial parts, roots	[40]
	*A. vulgaris*	aerial parts	[31,37,38,39]
**25.** Orientin	*A. hirtipedicellata* *A. procerrima* *A. sericata* *A. stricta*	leaves	[22]
	*A. armeniaca* *A. cimilensis* *A. hirsutiflora* *A. ikizdereensis* *A. orduensis*	aerial parts	[23]
**26.** Cynaroside	*A. mollis*	aerial parts	[36]
	*A. speciosa*	aerial parts	[21]
	*A. vulgaris*	aerial parts	[38,39]
**27.** Scolymoside	*A. speciosa*	aerial parts	[21]
**28.** Isorhamnetin	*A. acutiloba*	aerial parts, roots	[40]
**29.** Isorhamnetin-3-glucoside	*A. acutiloba*	aerial parts	[40]
**30.** Narcissoside	*A. acutiloba*	aerial parts, roots	[40]
**31.** Apigenin	*A. caucasica*	aerial parts	[10]
	*A. vulgaris*	aerial parts leaves	[37,39]
**32.** Vitexin	*A. hirtipedicellata* *A. procerrima* *A. sericata* *A. stricta*	leaves	[22]
	*A. armeniaca* *A. erzincanensis* *A. ikizdereensis* *A. orduensis*	aerial parts	[23]
**33.** Cosmosiin	*A. mollis*	aerial parts	[36]
	*A. vulgaris*	aerial parts	[38,39]
		leaves	[37]
**34.** Catechin	*A. barbatiflora*	aerial parts	[25]
	*A. caucasica*	aerial parts	[10]
	*A. glabra*	aerial parts	[32]
	*A. mollis*	aerial parts	[36]
	*A. persica*	aerial parts	[6]
	*A. vulgaris*	aerial parts leaves roots	[35,37,39]
**35.** Epicatechin	*A. glabra*	aerial parts	[32]
	*A. mollis*	aerial parts	[36]
	*A. persica*	aerial parts	[6]
	*A. vulgaris*	aerial parts leaves	[31,37]
**36.** Rhodiolgin	*A. mollis*	aerial parts	[27]
**37.** Gossypetin-3-*O-β*-D-galactopyranosyl-7-*O*-*α*-L-rhamnopyranoside	*A. mollis*	aerial parts	[27]
**38.** Myricetin	*A. vulgaris*	aerial parts	[31]
**39.** Genistein	*A. vulgaris*	aerial parts	[31,39]
**40.** Daidzein	*A. vulgaris*	aerial parts	[31]
**41.** Morin	*A. vulgaris*	aerial parts	[38]
**42.** Chrysoeriol	*A. vulgaris*	aerial parts	[39]
**43.** Naringenin	*A. vulgaris*	aerial partsleaves	[37,39]
**44.** Eriodictyol	*A. acutiloba*	aerial parts	[40]
**45.** Sinocrassoside D2	*A. mollis*	aerial parts	[27]
**46.** Luteolin 6-arabinose 8-glucose	*A. vulgaris*	leaves	[37]
**47.** Luteolin 6-glucose 8-arabinose	*A. vulgaris*	leaves	[37]
**48.** Apigenin 6-arabinose 8-galactose	*A. vulgaris*	leaves	[37]
**49.** Apigenin 6-rhamnose 8-glucose	*A. vulgaris*	leaves	[37]
**50.** Apigenin 7-*O*-neohesperidoside	*A. vulgaris*	leaves	[37]
**51.** Kaempferol 3,7-dirhamoside	*A. vulgaris*	leaves	[37]
**52.** Hesperetin	*A. vulgaris*	leaves	[37]
**53.** Kaempferol 3-(2-*p*-comaroyl)glucose	*A. vulgaris*	leaves	[37]
**54.** Rhamnetin	*A. vulgaris*	leaves	[37]
**55.** Aromadendrin glucoside	*A. persica*	aerial parts	[6]
**56.** Avicularin	*A. vulgaris*	aerial parts	[35]
**57.** Acacetin	*A. vulgaris*	leaves	[37]
**58.** Agrimoniin	*A. mollis*	leaves	[34]
	*A. persica*	aerial parts	[6]
	*A. viridiflora*	aerial parts	[41]
	*A. vulgaris*	leaves	[34]
	*A. xanthochlora*	aerial parts	[42]
**59.** Pedunculagin	*A. mollis*	leaves	[34]
	*A. persica*	aerial parts	[6]
	*A. vulgaris*	leaves	[34]
	*A. viridiflora*	aerial parts	[41]
	*A. xanthochlora*	aerial parts	[42]
**60.** Laevigatin F	*A. xanthochlora*	aerial part	[42]
**61.** Castalagin/vescalagin isomer	*A. mollis*	leaves	[34]
	*A. vulgaris*	leaves	[34]
**62.** Galloyl-HHDP hexose	*A. mollis*	leaves	[34]
	*A. persica*	aerial parts	[6]
	*A. vulgaris*	leaves	[34]
**63.** Trigalloyl hexose	*A. mollis*	leaves	[34]
**64.** Sanguiin	*A. mollis*	leaves	[34]
	*A. persica*	aerial parts	[6]
	*A. viridiflora*	aerial parts	[41]
	*A. vulgaris*	leaves	[34]
**65.** Methyl gallate	*A. mollis*	aerial parts	[36]
	*A. persica*	aerial parts	[6]
**66.** Casuarictin	*A. persica*	aerial parts	[6]
**67.** Digalloyl-galloyl galloside	*A. persica*	aerial parts	[6]
**68.** HHDP-hexoside	*A. viridiflora*	aerial parts	[41]
**69.** Brevifolincarboxylic acid	*A. viridiflora*	aerial parts	[41]
**70.** Tellimagrandin I	*A. viridiflora*	aerial parts	[41]
**71.** Tellimagrandin II	*A. viridiflora*	aerial parts	[41]
**72.** Benzoic acid	*A. vulgaris*	leaves	[37,43]
	*A. jumrukczalica*	leaves	[43]
**73.** Caffeic acid	*A. acutiloba*	aerial parts, roots	[40]
	*A. glabra*	aerial parts	[32]
	*A. jumrukczalica*	leaves	[43]
	*A. mollis*	aerial parts	[36]
	*A. vulgaris*	aerial parts leaves	[37,39,43]
**74.** Chlorogenic acid	*A. glabra*	aerial parts	[32]
	*A. mollis*	leaves	[34]
	*A. persica*	aerial parts	[6]
	*A. vulgaris*	aerial parts leaves	[31,34,37,39]
**75.** 2,5-Dihydroxybenzoic acid	*A. vulgaris*	aerial parts	[39]
**76.** 3,4-Dihydroxybenzoic acid	*A. glabra*	aerial parts	[32]
**77.** Ellagic acid	*A. mollis*	leaves	[34]
	*A. persica*	aerial parts	[6]
	*A. vulgaris*	aerial parts leaves	[31,38,44] [31,34,37]
**78.** Ferulic acid	*A. acutiloba*	aerial parts	[40]
	*A. vulgaris*	aerial parts leaves	[37,38,39]
**79.** Gallic acid	*A. acutiloba*	aerial parts roots	[40]
	*A. glabra*	aerial parts	[32]
	*A. jumrukczalica*	leaves	[43]
	*A. mollis*	aerial parts leaves	[34,36]
	*A. persica*	aerial parts	[6]
	*A. vulgaris*	aerial parts leaves roots	[31,34,35] [37,39,43] [45]
**80.** Gentisic acid	*A. acutiloba*	aerial parts, roots	[40]
	*A. jumrukczalica*	leaves	[43]
	*A. mollis*	aerial parts	[36]
	*A. vulgaris*	leaves	[43]
**81.** Protocatechuic acid	*A. acutiloba*	aerial parts, roots	[40]
	*A. jumrukczalica*	leaves	[43]
	*A. vulgaris*	aerial parts leaves	[37,39,43]
**82.** *p*-Coumaric acid	*A. acutiloba*	aerial parts roots	[40]
	*A. jumrukczalica*	leaves	[43]
	*A. vulgaris*	aerial parts leaves	[31,37,39,43]
**83.** 4-Hydroxybenzoic acid	*A. acutiloba*	aerial parts roots	[40]
	*A. jumrukczalica*	leaves	[43]
	*A. vulgaris*	aerial parts leaves	[37,39,43]
**84.** Mandelic acid	*A. jumrukczalica*	leaves	[43]
	*A. vulgaris*	leaves	[43]
**85.** 3,4,5-Methoxycinnamic acid	*A. vulgaris*	leaves	[37]
**86.** *β*-Resorcylic acid	*A. jumrukczalica*	leaves	[43]
	*A. vulgaris*	leaves	[43]
**87.** Rosmarinic acid	*A. acutiloba*	aerial parts roots	[40]
	*A. vulgaris*	aerial parts leaves	[31,37]
**88.** Salicylic acid	*A. acutiloba*	aerial parts roots	[40]
	*A. jumrukczalica*	leaves	[43]
	*A. vulgaris*	leaves	[37,43]
**89.** Sinapic acid	*A. jumrukczalica*	leaves	[43]
	*A. vulgaris*	aerial parts leaves	[31,43]
**90.** Syringic acid	*A. acutiloba*	aerial parts roots	[40]
	*A. jumrukczalica*	leaves	[43]
	*A. vulgaris*	leaves	[43]
**91.** *Trans*-cinnamic acid	*A. jumrukczalica*	leaves	[43]
	*A. vulgaris*	leaves	[37,43]
**92.** 3,4,5-Trimethoxymandelic acid	*A. jumrukczalica*	leaves	[43]
	*A. vulgaris*	leaves	[43]
**93.** Vanillic acid	*A. acutiloba*	aerial parts roots	[40]
	*A. jumrukczalica*	leaves	[43]
	*A. vulgaris*	leaves	[37,43]
**94.** Quinic acid	*A. vulgaris*	aerial parts	[39]

**Table 8 plants-11-02709-t008:** Hydroxycinnamic acid derivatives.

Compound	R_1_	R_2_	R_3_	R_4_	R_5_	MW
**73**	H	OH	OH	H	H	180
**78**	H	OCH_3_	H	OH	H	194
**82**	H	H	OH	H	H	164
**85**	H	OCH_3_	OCH_3_	OCH_3_	H	238
**89**	H	OCH_3_	OH	OCH_3_	H	224
**91**	H	H	H	H	H	148

**Table 9 plants-11-02709-t009:** Benzoic acid derivatives.

Compound	R_1_	R_2_	R_3_	R_4_	R_5_	MW
**72**	H	H	H	H	H	122
**79**	H	OH	OH	OH	H	170
**80**	OH	H	H	OH	H	154
**81**	H	OH	OH	H	H	154
**83**	H	H	OH	H	H	138
**86**	OH	H	OH	H	H	154
**88**	OH	H	H	H	H	138
**90**	OH	OCH_3_	OH	OCH_3_	H	198
**93**	H	OCH_3_	OH	H	H	168

**Table 10 plants-11-02709-t010:** The overview on the antioxidant activities in *Alchemilla* species.

Species	Plant Part/Extract	Antioxidant Assay	Antioxidant Effect	References
*A. acutiloba*	aerial parts, 60% methanol	DPPH	IC_50_ = 18.69 µg/mL DE	[40]
	aerial parts, 60% methanol	ABTS	IC_50_ = 6.17 µg/mL DE	
	aerial parts, 60% methanol	CHEL	IC_50_ = 21.60 µg/mL DE	
	roots, 60% methanol	DPPH	IC_50_ = 29.87 µg/mL DE	
	roots, 60% methanol	ABTS	IC_50_ = 14.29 µg/mL DE	
	roots, 60% methanol	CHEL	IC_50_ = 25.76 µg/mL DE	
	aerial parts, butanol fraction	DPPH	IC_50_ = 8.96 µg/mL DE	
	aerial parts, butanol fraction	ABTS	IC_50_ = 1.42 µg/mL DE	
	aerial parts, butanol fraction	CHEL	IC_50_ = 11.43 µg/mL DE	
	roots, butanol fraction	DPPH	IC_50_ = 12.08 µg/mL DE	
	roots, butanol fraction	ABTS	IC_50_ = 8.78 µg/mL DE	
	roots, butanol fraction	CHEL	IC_50_ = 12.33 µg/mL DE	
	aerial parts, diethyl acetate fraction	DPPH	IC_50_ = 8.83 µg/mL DE	
	aerial parts, diethyl acetate fraction	ABTS	IC_50_ = 6.54 µg/mL DE	
	aerial parts, diethyl acetate fraction	CHEL	IC_50_ = 18.89 µg/mL DE	
	roots, diethyl acetate fraction	DPPH	IC_50_ = 15.37 µg/mL DE	
	roots, diethyl acetate fraction	ABTS	IC_50_ =10.39 µg/mL DE	
	roots, diethyl acetate fraction	CHEL	IC_50_ = 19.30 µg/mL DE	
	aerial parts, diethyl ether fraction	DPPH	IC_50_ = 41.46 µg/mL DE	
	aerial parts, diethyl ether fraction	ABTS	IC_50_ = 16.28 µg/mL DE	
	aerial parts, diethyl ether fraction	CHEL	IC_50_ = 25.51 µg/mL DE	
	roots, diethyl ether fraction	DPPH	IC_50_ = 51.42 µg/mL DE	
	roots, diethyl ether fraction	ABTS	IC_50_ = 24.82 µg/mL DE	
	roots, diethyl ether fraction	CHEL	IC_50_ = 44.12 µg/mL DE	
*A. alpina*	aerial parts, methanol	DPPH	% Inhibition = 45.4–94.4%	[47]
*A. arvensis*	leaves, methanol	DPPH	IC_50_ = 97.72 µg/mL	[48]
	leaves, hexane		IC_50_ = 11.22 µg/mL	
	leaves, acetone		IC_50_ = 4.86 µg/mL	
*A. barbatiflora*	aerial parts, methanol	DPPH	% Inhibition = 83.44–95.35%	[25]
	aerial parts, hexane fraction		% Inhibition = 18.6–59.62%	
	aerial parts, chloroform fraction		% Inhibition = 67.17–91.11%	
	aerial parts, water fraction		% Inhibition = 83.06–97.17%	
	aerial parts, methanol	SOD	% Inhibition = 83.34–85.83%	
	aerial parts, hexane fraction		% Inhibition = 9.80%	
	aerial parts, chloroform fraction		% Inhibition = 12.84–42.73%	
	aerial parts, water fraction		% Inhibition = 81.07%	
	aerial parts, methanol	PRA	Absorbance 0.932–1.280	
	aerial parts, hexane fraction		Absorbance 0.355–0.612	
	aerial parts, chloroform fraction		Absorbance 0.640–0.820	
	aerial parts, water fraction		Absorbance 1.158–1.516	
	aerial parts, methanol	FRAP	44.32 mg BHAE/g DE	
	aerial parts, chloroform fraction		15.76 mg BHAE/g DE	
	aerial parts, water fraction		93.46 mg BHAE/g DE	
*A. bulgarica*	aerial parts, 80% methanol	DPPH	IC_50_ = 75.63 µg/mL	[49]
*A. crinita*	aerial parts, 80% methanol	DPPH	IC_50_ = 46.03 µg/mL	[49]
*A. ellenbergiana*	aerial parts, hexane	DPPH	IC_50_ = 7.1 µg/mL	[50]
*A. ellenbergiana*	aerial parts, ethanol	DPPH	IC_50_ = 243.6 µg/mL	[51]
	aerial parts, methanol		IC_50_ = 243.1 µg/mL	
*A. erythropoda*	aerial parts, 80% methanol	DPPH	IC_50_ = 30.67 µg/mL	[50]
*A. glabra*	aerial parts, 80% acetone in 0.2% formic acid	ORAC	IC_50_ = 1337 μmol TE/g	[32]
		TRAP	IC_50_ = 1815 μmol TE/g	
		HORAC	IC_50_ = 1999 μmol GAE/g	
*A. glabra*	aerial parts, 80% methanol	DPPH	IC_50_ = 34.89 µg/mL	[49]
*A. glaucescens*	aerial parts, 80% methanol	DPPH	IC_50_ = 36.10 µg/mL	[49]
*A. jumrukczalica*	leaves, 80% methanol	DPPH	IC_50_ = 12.09 µg/mL	[43]
*A. mollis*	shoots grown in vitro on different nutrient media	DPPH	IC_50_ = 18.6–38.1 μg/mL	[52]
	leaves of ex vitro adapted plants in Bulgarian mountains Vitosha		IC_50_ = 13.1 μg/mL	
	one year old in vivo plantsgrown in Bulgarian mountains Viotsha		IC_50_ = 27.5 μg/mL	
	one year old in vivo plantsgrown in Bulgarian mountains Rhodopes		IC_50_ = 22.2 μg/mL	
*A. mollis*	leaves, 50% ethanol	DPPH	IC_50_ = 42.4 μg/mL	[53]
		ABTS	IC_50_ = 7.8 μg/mL	
*A. mollis*	aerial parts, water	DPPH	IC_50_ = 0.264 mg/mL	[54]
	aerial parts, deodorized water		IC_50_ = 0.146 mg/mL	
	aerial parts, 50% methanol		IC_50_ = 0.161 mg/mL	
	aerial parts, water	ABTS	0.90 mmol/L/Trolox	
	aerial parts,deodorized water		0.4 mmol/L/Trolox	
	aerial parts,50% methanol		0.4 mmol/L/Trolox	
*A. mollis*	aerial parts, 70% methanol	DPPH	IC_50_ = 0.21 mg/mL	[36]
	aerial parts,water		IC_50_ = 0.24 mg/mL	
	aerial parts, 70% methanol	ABTS	TEAC = 0.75 mmol/Trolox	
	aerial parts,water		TEAC = 0.83 mmol/Trolox	
	aerial parts,hexane, ethyl acetate, methanol, butanol, 70% methanol, water	Inhibition of β-carotene/linoleic acid co-oxidation	no data	
*A. mollis*	dry stalks, aqueous ethanol	FRAP	TEAC = 382.78 mmol TE/g DW	[55]
	dry stalks, aqueous ethanol	CUPRAC	TEAC = 363.79 mmol TE/g DW	
	dry stalks, aqueous ethanol	DPPH	TEAC = 247.58 mmol TE/g DW	
	dry stalks, aqueous ethanol	ABTS	TEAC = 308.44 mmol TE/g DW	
*A. mollis*	aerial parts, methanol	DPPH	IC_50_ = 31.7 μg/mL	[27]
	aerial parts, ethyl acetate fraction		IC_50_ = 9.8 μg/mL	
	aerial parts, petroleum fraction		IC_50_ = > 200 μg/mL	
	aerial parts, chloroform fraction		IC_50_ = > 200 μg/mL	
	aerial parts, water residue fraction		IC_50_ = 42.5 μg/mL	
*A. persica*	aerial parts, 80% methanol	DPPH	IC_50_ = 0.055 M	[3]
	roots, 80% methanol		IC_50_ = 0.151 M	
	aerial parts, 80% methanol	TBARS	MDA = 5.9 nmol/mL	
	roots, 80% methanol		MDA = 19.08 nmol/mL	
*A. monticola*	aerial parts, 80% methanol	DPPH	IC_50_ = 32.72 μg/mL	[49]
A. obtusa	aerial parts, 80% methanol	DPPH	IC_50_ = 26.35 μg/mL	[49]
*A. sericata*	aerial parts, hexane	DPPH	IC_50_ = 185 μg/mL	[56]
*A. vulgaris*	leaves, 50% ethanol	DPPH	% Inhibition = 71.8%	[57]
*A. vulgaris*	aerial parts, methanol	DPPH	IC_50_ = 5.40 µg/mL	[58]
		ABTS	IC_50_ = 60.10 µg/mL	
*A. vulgaris*	aerial parts, methanol	DPPH	IC_50_ = 5.96 µg/mL	[11]
	roots, methanol		IC_50_ = 11.86 µg/mL	
	aerial parts, methanol	ABTS	IC_50_ = 14.80 µg/mL	
	roots, methanol		IC_50_ = 32.49 µg/mL	
	aerial parts, methanol	Hydroxyl radical scavenging activity	IC_50_ = 13.06 µg/mL	
	roots, methanol		IC_50_ = 18.44 µg/mL	
	aerial parts, methanol	Inhibition of lipid peroxidation	IC_50_ = 31.91 µg/mL	
	roots, methanol		IC_50_ = 475.13 µg/mL	
	aerial parts, methanol	Reducing power	IC_50_ = 632.99 mg TE/g DE	
	roots, methanol		IC_50_ = 607.52 mg TE/g DE	
	aerial parts, methanol	Total antioxidant activity	IC_50_ = 265.62 mg AA/g DE	
	roots, methanol		IC_50_ = 316.47 mg AA/g DE	
*A. vulgaris*	leaves, 80% ethanol	DPPH	% inhibition = 131.74%	[37]
*A. vulgaris*	roots, 50% ethanol	TEAC	68.21 mmol TE/g DW	[59]
		FRAP	40.12 mmol TE/g DW	
*A. vulgaris*	aerial parts, cyclohexane	DPPH	IC_50_ = 23.12 µg/mL	[44]
*A. vulgaris*	aerial parts, 80% methanol	DPPH	153.30 mg TE/g DE	[39]
		ABTS	143.55 mg TE/g DE	
		CUPRAC	216.14 mg TE/g DE	
		PRAP	1.77 mmol TE/g DE	
		CHEL	42.58 mg EDTAE/g DE	
		FRAP	7899.45 mg AAE/g DE	
	aerial parts, 70% ethanol	DPPH	95.99 mg TE/g DE	
		ABTS	119.62 mg TE/g DE	
		CUPRAC	203.53 mg TE/g DE	
		PRAP	1.57 mmol TE/g of DE	
		CHEL	42.32 mg EDTAE/g DE	
		FRAP	6405.75 mg AAE/g DE	
	aerial parts, 70% ethyl-acetate	DPPH	502.56 mg TE/g DE	
		ABTS	174.05 mg TE/g DE	
		CUPRAC	283.16 mg TE/g DE	
		PRAP	2.22 mmol TE/g DE	
		CHEL	37.96 mg EDTAE/g DE	
		FRAP	8745.31 AAE/g DE	
	aerial parts, water	DPPH	89.25 mg TE/g DE	
		ABTS	37.50 mg TE/g DE	
		CUPRAC	78.56 mg TE/g DE	
		PRAP	0.53 mmol TE/g DE	
		CHEL	39.23 mg EDTAE/g DE	
		FRAP	3240.09 mg AAE/g DE	
*A. vulgaris*	aerial parts, ethanol	DPPH	IC_50_ = 0.11 μg/mL	[38]
	aerial parts, water		IC_50_ = 27.22 μg/mL	
	aerial parts, propylene glycolic		IC_50_ = 2.88 μL/mL	
*A. vulgaris*	aerial parts, 70% ethanol	DPPH	87.95% (at 3 mg/mL) and 80.71% (at 1.5 mg/mL)	[31]
*A. vulgaris*	leaves, 80% methanol	DPPH	IC_50_ = 19.62 µg/mL	[43]
*A. xanthochlora*	aerial parts, 80% methanol	DPPH	IC_50_ = 41.78 µg/mL	[49]
*A. xanthochlora*	leaves,hexane	TLC-DPPH analysis, DPPH	no data	[13]
	leaves, chloroform		no data	
	leaves, ethylacetate		no data	
	leaves, methanol		no data	
	leaves, water		no data	

DPPH, 2.2-diphenyl-1-picryl-hydrazyl free radical scavenging activity; ABTS, 2,2′-azinobis [3-ethylbenzthiazoline]-6-sulfonic acid decolorization assay; CHEL, metal chelating activity; DE, dry extract; SOD, superoxide radical scavenging; PRAP, phosphomolibdenum-reducing antioxidant power; FRAP, ferric-reducing antioxidant power; BHAE, butylated hydroxyanisole equivalents; ORAC, oxygen radical absorbance capacity; TE, Trolox equivalents; TRAP, total peroxyl-radical antioxidant parameter; HORAC, hydroxyl radical averting capacity; GAE, gallic acid equivalents; CUPRAC, cupric reducing antioxidant capacity; DW, dry weight; TBARS, thiobarbituric acid reactive substances assay; MDA, malondialdehyde level; TEAC, Trolox equivalent antioxidant capacity; EDTAE, ethylenediaminetetraacetic acid equivalent; AAE, ascorbic acid equivalents; no data—results on the figures, without values.

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
