# Peer review of "Phenolic Composition and Antioxidant Activity of Alchemilla Species"

_plants, 2022, doi:10.3390/plants11202709_

Round 1

Reviewer 1 Report

Sebastian Kanak and co-authors  focus on the phenolic compounds composition and antioxidative activity of Alchemilla species. This work is very interesting.  The respective suggestions are as follows:

1. It is recommended that the author provide representative images of Alchemilla species

2. It is advised to  list more  information  of antioxidant activities, such as the methods, etc.

3. Molecular formula of representative polyphenols should be provided.

4. The possible health effects of Alchemilla species polyphenols?

Reviewer 2 Report

In this review article, Dos Santos and his fellows summarized the published literature till June 2022 discussing Alchemilla species and concerning antioxidant activities. The review article sounds scientifically interesting and to the best of the available knowledge, this topic is not previously reviewed elsewhere.

The review article in its present form may not be possible for publication in Plants Journal unless some important points bein addressed as follow:

1- The whole English language of the review article needs to be edited and my advice is to seek help from the Publisher's editing services and to ask one of the peers who is native English speaker to read the whole review article for language and punctuation adjustment.

2- The Introduction part is too long covering unnecessary historical points. This part needs to be more precisely defined and clearly addressing the aim of the review article.

3- The review article mentioned up to 96 compounds listed in Table 1 and also within the text with bold numbers, however none of their chemical structures were drawn in the article. It is important to insert the chemical structures of all mentioned compounds for the clearness and the convenience to the readers.

4- Table 1 needs to be adjusted carefully for the alignment and the cell size to fit to the text within.

5- The references are mentioned within the context in numerical values, but the list at the end is not fitting to this style, e.g. References 23 and 24 mentioned the publication year as 2012a and 2012b, respectively. This is not going with the reference list instructions of Plants Journal.

6- In conclusion, the review article in its current status is difficult to be accepted for publication in Plants Journal.
The reviewer strongly recommends major revisions to the current version of the review article and afterwards the publication can be considered.

Author Response

Dear Reviewer,

We enclose the revised version of our manuscript. We are thankful to the reviewer for reviewing and giving valuable suggestions to further improve our manuscript. We have carefully gone over the varies queries raised and have tried to address them. We hope that the reviewer will be satisfied with the changes that have been incorporated.

The corrections made are highlighted in yellow colour within the text of resubmitted manuscript.

1. The whole English language of the review article needs to be edited and my advice is to seek help from the Publisher's editing services and to ask one of the peers who is native English speaker to read the whole review article for language and punctuation adjustment.

We have corrected English language

2. The Introduction part is too long covering unnecessary historical points. This part needs to be more precisely defined and clearly addressing the aim of the review article.

We have corrected this.

3. The review article mentioned up to 96 compounds listed in Table 1 and also within the text with bold numbers, however none of their chemical structures were drawn in the article. It is important to insert the chemical structures of all mentioned compounds for the clearness and the convenience to the readers..

We have added chemical structures.

4. Table 1 needs to be adjusted carefully for the alignment and the cell size to fit to the text within.

We have corrected this.

5. The references are mentioned within the context in numerical values, but the list at the end is not fitting to this style, e.g. References 23 and 24 mentioned the publication year as 2012a and 2012b, respectively. This is not going with the reference list instructions of Plants Journal.

We have corrected this.

 Sincerely Yours,

Authors

Reviewer 3 Report

I believe that the work presented for review is of a high technical level, but it requires substantive amendments (please post new items). The research and the prepared article are of a very high substantive level, based only on the latest scientific reports from the last few years.

Specific comments:

1. I strongly suggest authors to introduce more keywords. The usefulness of keywords is to make the article both more and more easily searchable visible after its publication through commonly used search engines.

2. Abstract and introduction are interesting, but in my opinion, it donot fully cover the topic.

Round 2

Reviewer 1 Report

The revised manuscript should be published.